Group B Streptococcus and the vaginal microbiome among pregnant women: a systematic review

Lim Sungju 1
Rajagopal Shilpa 2
Jeong Ye Ryn 1
Nzegwu Dumebi 3
Wright Michelle L. michelle.wright@utexas.edu 1 4
1 School of Nursing, The University of Texas at Austin , Austin , TX , United States of America
2 College of Natural Sciences, Biology Instructional Office, The University of Texas at Austin , Austin , TX , United States of America
3 College of Liberal Arts, Department of Health and Society, The University of Texas at Austin , Austin , TX , United States of America
4 Dell Medical School, Department of Women’s Health, University of Texas at Austin , Austin , TX , United States of America
Kurita Takeshi
Electronic publication date: 2021 May 17
Publication date: 2021
Volume: 9
Electronic Location ID: e11437
Received 2021 Jan 25; Accepted 2021 Apr 20
Copyright: ©2021 Lim et al.
Copyright year: 2021
Copyright holder: Lim et al.
License: This is an open access article distributed under the terms of the Creative Commons Attribution License, which permits unrestricted use, distribution, reproduction and adaptation in any medium and for any purpose provided that it is properly attributed. For attribution, the original author(s), title, publication source (PeerJ) and either DOI or URL of the article must be cited.
License URL: https://creativecommons.org/licenses/by/4.0/

Keywords: Vaginal microbiome, Pregnancy, 16S ribosomal RNA, Whole genome sequencing, Streptococcus agalactiae, Group B Streptococcus, GBS

Funding: National Institute of Nursing Research K01NR017903 Michelle L. Wright is supported by the National Institute of Nursing Research (K01NR017903). The funders had no role in study design, data collection and analysis, decision to publish, or preparation of the manuscript.

==============================
Background

Vaginal microbiome studies frequently report diversity metrics and communities of microbiomes associated with reproductive health outcomes. Reports of Streptococcus agalactiae (also known as Group B Streptococcus or GBS), the leading cause of neonatal infectious morbidity and mortality, are notably lacking from the studies of the vaginal microbiome, despite being a known contributor to preterm birth and other complications. Therefore, the purpose of this systematic review was to explore the frequency of GBS reporting in vaginal microbiome literature pertaining to pregnancy and to examine methodological bias that contributes to differences in species and genus-level microbiome reporting. Lack of identification of GBS via sequencing-based approaches due to methodologic or reporting bias may result incomplete understanding of bacterial composition during pregnancy and subsequent birth outcomes.

Methodology

A systematic review was conducted following the PRISMA guideline. Three databases (PubMed, CINAHL, and Web of Science) were used to identify papers for review based on the search terms “vaginal microbiome”, “pregnancy”, and “16S rRNA sequencing”. Articles were evaluated for methods of DNA extraction and sequencing, 16S region, taxonomy classification database, number of participants or vaginal specimens, and pregnancy trimester.

Results

Forty-five research articles reported employing a metagenomic approach or 16S approach for vaginal microbiome analysis during pregnancy that explicitly reported taxonomic composition and were included in this review. Less than 30% of articles reported the presence of GBS (N = 13). No significant differences in methodology were identified between articles that reported versus did not report GBS. However, there was large variability across research methods used for vaginal microbiome analysis and species-level bacterial community reporting.

Conclusion

Considerable differences in study design and data formatting methods may contribute to underrepresentation of GBS, and other known pathogens, in existing vaginal microbiome literature. Previous studies have identified considerable variation in methodology across vaginal microbiome studies. This study adds to this body of work because in addition to laboratory or statistical methods, how results and data are shared (e.g., only analyzing genus level data or 20 most abundant microbes), may hinder reproducibility and limit our understanding of the influence of less abundant microbes. Sharing detailed methods, analysis code, and raw data may improve reproducibility and ability to more accurately compare microbial communities across studies.

Introduction

Variations within the vaginal microbiome have been associated with higher risk for preterm birth and other pregnancy complications, such as chorioamnionitis. Microbes associated with these complications such as Streptococcus agalactiae and Gardnerella vaginalis, are frequently present in polymicrobial, as well as Lactobacillus predominant vaginal microbiomes (Brooks et al., 2017). In light of these considerations, a growing body of research has focused on characterizing the vaginal microbiome using metagenomic approaches. Among the most common methodologies are those that rely on universal targets, 16S ribosomal RNA and cpn60 genes, allowing for more detailed taxonomic classification and profiling of the vaginal microbiome during pregnancy than culture based or targeted sequencing methods.

However, data from amplicon-based metagenomic characterizations suggest that such methods provide an incomplete understanding of the species-level diversity present in the vaginal microbiome (Bayar et al., 2020). For example, there is a notable lack of discussion of S. agalactiae, commonly referred to as Group B Streptococcus (GBS), a known pathogen associated with poor maternal, fetal, and neonatal outcomes in amplicon-based metagenomic vaginal microbiome studies during pregnancy. To date, many vaginal microbiome papers related to birth outcomes primarily focus on differences in community state types, which is largely determined by the abundance and type of Lactobacillus present (Brooks et al., 2017). Focus only on the larger vaginal community state type for analysis may limit evaluation and reporting of microbial species that are known pathogens of pregnancy that are present in low abundance.

Currently, in the United States, screening for GBS occurs between 36 to 38 weeks of pregancy (Committee on Obstetric Practice, 2020), with those who test positive receiving antibiotic treatment during labor to help prevent early-onset neonatal GBS infection. Previous data indicate that approximately one in five women are colonized with GBS (Russell et al., 2017). Yet, apart from this routine clinical screening near the time of birth, there are limited metagenomic studies that identify or evaluate GBS during pregnancy. While screening for GBS prior to birth has reduced early onset neonatal sepsis caused by GBS in the US (Phares et al., 2008), the transient nature of maternal GBS colonization is not well understood (McCoy et al., 2020; Hansen et al., 2004). Improved understanding of the dynamics of maternal GBS colonization during pregnancy can provide additional insights for preventing GBS complications affecting maternal and fetal health throughout pregnancy and birth (e.g., miscarriage and stillbirth).

To our knowledge, there is no systematic review addressing lack of representation of known pathogens, such as GBS, associated with pregnancy complications in existing vaginal microbiome literature. Therefore, this systematic review was conducted to: (1) determine how often GBS is reported in studies of the vaginal microbiome during pregnancy, and (2) analyze the study methods used to identify potential methodological bias in GBS reporting or underrepresentation in vaginal microbiome studies during pregnancy.

Methods

Protocol and registration

This systematic review was conducted in accordance with the Preferred Reporting Items for Systematic reviews and Meta-analyses (PRISMA) guidelines (Moher et al., 2009).

Eligibility criteria

Original studies that investigated the vaginal microbiome during human pregnancy, written in English, and published within the past 10 years were included. Only studies with pregnant women as study participants and reported microbiome taxonomy were included. Review papers, editorials, commentaries, and methodological papers including procedure/protocols were excluded. Research studies that used a targeted approach (i.e.: only include specific microbial taxa, such as Lactobacillus species, were analyzed via microbe specific PCR), were also excluded.

Information sources

Three electronic databases PubMed, CINAHL, and Web of Science were used for data extraction (date last searched: June 30, 2020).

Search

We searched these databases using two groups of search terms: (“vaginal microbiome” or “vaginal microbiota” or “pregnancy”) and (“16S ribosomal RNA” or “16S rRNA” or “whole genome sequencing”). For example, in PubMed, ((((vaginal microbiome) OR (vaginal microbiota))) AND pregnancy) AND ((16s ribosomal rRNA) OR (16s rRNA) OR (whole genome sequencing) OR (metagenomic)) were entered in the search window with the display option ‘best match’. Limitations of publication date as ‘10 years’, species as ‘humans’, and language as ‘English’ were used (Appendix S1).

Study selection

After combining articles from the three databases and removing duplicates, five reviewers (SL, SR, YJ, DN, MLW) independently screened all results with titles and abstracts. Disagreements were resolved by discussion among all five reviewers to reach consensus. After the screening, eligibility was checked with full text by five reviewers (SL, SR, YJ, DN, MLW), who each reviewed one fifth of the total articles and cross-reviewed another one fifth. Disagreements were resolved by discussion to reach consensus.

Data collection process

Data was extracted from the included studies by using a collaborative Google Sheets document. Each of five reviewers (SL, SR, YJ, DN, MLW) extracted one fifth of the included studies and cross-checked the accuracy of extracted data from another one fifth. Ambiguity was resolved through discussion among all reviewers.

Data items

The initial data extraction tool contained the items as follows: study title, study purpose, whether GBS was reported, level of analysis - genus/species (in the case of GBS not reported), sequencing methods, 16S rRNA primer region (for studies using 16S rRNA sequencing), sequencing platform, taxonomy classification database, DNA collection kit, DNA extraction kit, number of pregnant women, number of vaginal specimens, age of pregnant women, pregnancy period in which the specimen collected, ethnicity/region of the participants/study design. In studies that reported taxonomy to the species level and GBS was not detected, we also assessed the articles to determine if Streptococcus at the genus level was reported.

Risk of bias in individual studies

No previous studies have exclusively examined the methodology of vaginal microbiome studies related to detection of known common opportunistic pathogens, like GBS. One of the main purposes of this review was to evaluate if methodological differences/biases in vaginal microbiome studies contribute to reporting or underrepresentation of GBS in existing microbiome literature.

Summary measures

Our primary outcome was to identify the number of studies that reported the presence of GBS within the vaginal microbiome. Due to methodological heterogeneity, the differences across analytic and sequencing methodologies between studies that reported versus did not report GBS were compared narratively.

Synthesis of results

After data extraction, the included studies were divided into two groups: GBS reported and GBS not reported. Independent t-tests were conducted to compare the number of pregnant women and the number of vaginal specimens between GBS reported and not reported groups. The commonalities and differences among 16S rRNA primer regions used were compared. We then qualitatively compared the commonalities and differences of the other methodological categories between two groups, since methods were too varied to compare statistically.

Results

Study selection

Our search strategy initially yielded 278 manuscripts (Fig. 1). After removing 68 duplicates, 210 article titles and abstracts were screened, and 159 studies were excluded because they did not meet inclusion criteria. Full text was reviewed for the remaining 52 manuscripts, and six additional studies were excluded for failing to meet inclusion criteria. A total of 45 studies were included and reviewed in this study.

Figure 1 The flow diagram of this study.

Study characteristics and results of individual studies

The characteristics of each study that reported (n = 13) (Fettweis et al., 2014; Ghartey et al., 2014; Romero et al., 2014a; Romero et al., 2014b; Bisanz et al., 2015; MacIntyre et al., 2015; Kindinger et al., 2017; Brown et al., 2018; Hočevar et al., 2019; Purkayastha et al., 2019; Romero et al., 2019; Tabatabaei et al., 2019; Al-Memar et al., 2020) or did not report GBS (n = 32) (Dominguez-Bello et al., 2010; Hernández-Rodríguez et al., 2011; Aagaard et al., 2012; Frank et al., 2012; Hyman et al., 2012; Hyman et al., 2014; Walther-António et al., 2014; Baldwin et al., 2015; DiGiulio et al., 2015; Huang et al., 2015; Brumbaugh et al., 2016; Jayaprakash et al., 2016; Lauder et al., 2016; Nelson et al., 2016; Subramaniam et al., 2016; Callahan et al., 2017; Freitas et al., 2017; Nasioudis et al., 2017b; Nasioudis et al., 2017a; Roesch et al., 2017; Stout et al., 2017; Goltsman et al., 2018; Leizer et al., 2018; Matsumoto et al., 2018; Wylie et al., 2018; Chen et al., 2019; Dobbler et al., 2019; He et al., 2019; Jefferson et al., 2019; Liu et al., 2019; Price et al., 2019; Witkin et al., 2019) are described in Tables S1 and S2 respectively.

Synthesis of results

Among the 45 studies reviewed, less than 30% of manuscripts reported the presence of GBS within the results, figures, or microbial taxonomy tables. To determine the frequency in which GBS was reported, and to compare analytic methods across studies, we analyzed: time of sample collection, number of research participants and samples, DNA extraction method, type of sequencing completed, and database used to assign taxonomy (Table 1).

Table 1 Comparison of GBS reported and GBS not reported studies.

	GBS Reported (n = 13)	GBS Not Reported (n = 32)	
Trimesters	1st trimester (1)
2nd trimester (0)
3rd trimester (3)
All trimesters (5)
2nd and 3rd (2)
Unspecified (2)	1st trimester (5)
2nd trimester (6)
3rd trimester (6)
All trimesters (12)
1st and 2nd (2)
2nd and 3rd (5)
Unspecified (1)	
Number of pregnant women	Mean: 111.154 (SD 121.817)
Range [8–450], Total # 1445	Mean: 82.594 (SD 91.331)
Range [6–461], Total # 2643	
Number of vaginal specimens	(11/13 studies; 2 unspecified)
Mean: 184
Standard deviation: 155.793
Range: [8–450]
Total #: around 2086	(32/32 studies)
Mean: 214
Standard deviation: 418.706
Range: [4–2179]
Total #: around 6848	
DNA extraction kit	QIAamp DNA Mini Kit (6)
MoBio PowerSoil Kit (3)
Custom (4)	QIAamp (6)
MoBio (13)
DNeasy (2)
Others (5)
Custom (5)	
Sequencing method	16S rRNA Sequencing (13)	16S rRNA Sequencing (29)
Shotgun sequencing (2)
Cpn60 (2)	
16S rRNA region	V1–V2 (4), V1–V3 (3), V3–V4 (1), V4 (3), V6 (1), full (1)	V1–V2 (2), V1–V3 (8), V2 (1), V3 (1), V3–V4 (3), V3–V5 (7), V4 (5), V4–V5 (1), V4–V6 (1), full (6)	

Participant and sample characteristics

There were no differences in the sample sizes (mean comparison of the number of specimens: t = 0.372, df = 43, p = 0.712; number of women: t = 0.862, df = 43, p = 0.394). However, there was large variability in sample size and number of specimens collected in studies that reported and did not report GBS. Additionally, four manuscripts that did not report GBS within the taxonomy analyzed, or overall metagenomic results, did report positive culture-based GBS clinical screening results for women within the study (Brumbaugh et al., 2016; Jayaprakash et al., 2016; Roesch et al., 2017; Leizer et al., 2018).

DNA extraction kits, Sequencing method, and Taxonomy classification

A variety of approaches were employed for DNA extraction; custom developed protocols, along with two standard extraction kits (Table 1) were used in studies reporting GBS within the vaginal microbiome. However, these kits and custom approaches were also used in studies that did not report GBS. In terms of sequencing methods, all studies reporting GBS used 16S rRNA sequencing, as well as most studies that did not report the presence of GBS. Additionally, there was no notable difference found in the 16S rRNA hypervariable region used across either set of studies. Similarly, multiple taxonomy databases were used for studies that did and did not report the presence of GBS.

Risk of bias across studies

The included studies were not originally conducted with a specific focus on GBS. However, most studies were evaluating the composition of the microbiome across pregnancy, or related to birth outcomes that GBS can influence, such as preterm birth (Tables S1 and S2). There is notable bias against GBS reporting when only genus level data are reported. And the practice of only speciating Lactobacillus in vaginal microbiome studies may need to be reconsidered, given GBS cannot be differentiated without also speciating Streptococcus. For example, seven studies that did not report GBS only assigned taxonomy to the genus level, 9 studies only assigned species level assignments for Lactobacillus spp., three studies only assigned species to Lactobacillus spp. and Gardnerella, and three studies that did assign taxonomy to species level for analysis did not report species level results (Table S1). Given that species other than Lactobacillus can negatively impact vaginal health and birth outcomes, these methods of analysis and reporting contribute to positive bias towards Lactobacillus reporting in the vaginal microbiome literature and negative bias towards other relevant pathogenic species that are less prevalent. Further, four studies only reported the top 15–21 most abundant species or taxa (Table S1). Since some vaginal pathogens present in low abundance can cause pregnancy complications the underrepresentation of these pathogens, such as GBS, may indicate bias against reporting of microbes that are not the most abundant taxonomic groups but still could be contributing to the poor outcomes being studied.

Discussion

In this review, we explored how often GBS was reported in studies of the vaginal microbiome during pregnancy. Methods related to GBS reporting were compared to identify potential methodological bias. Among 45 studies, 13 (28.9%) reported GBS. However, due to the heterogeneity of methods across study approaches, we were not able to detect any systematic differences that appeared to enhance or hinder GBS reporting.

Methodologic differences and sources of bias

There are known sources of bias in metagenomic studies associated with analysis methods (Committee on Obstetric Practice, 2020; Brooks et al., 2015; Knight et al., 2018). Brooks and colleagues determined that microbiome community composition can be biased at various steps including DNA extraction, PCR amplification, sequencing, and taxonomic classification (Brooks et al., 2015). The researchers reported that more than 85% of the microbiome community results were biased by small variations in methods (less than 5%). Specifically, it has been determined that variations in DNA extraction and amplification can result in decreased representation of Streptococcus species. If women with GBS colonization only harbor small amounts of the bacteria, it is possible that bias introduced by these steps could have contributed to the limited representation of GBS in vaginal microbiome data.

Such underrepresentation of known pathogens can significantly limit the clinical scope of vaginal microbiome studies. For example, many studies only reported results related to community state types or analyzed the most abundant species. Consequently, if thresholds for inclusion are based on sample abundance (e.g., 5% of sample or 5,000 reads), GBS data may have been excluded if it did not meet the threshold. Furthermore, selection of 16S regions for amplification and associated primers can influence genus and species level resolution (Graspeuntner et al., 2018). For example, it is generally more difficult to distinguish Lactobacillus species within the V4 region, but may be easier to do so in V1–V3 (Fettweis et al., 2012). Conversely, G. vaginalis is more difficult to detect and differentiate within the V1–V2 region because its sequence is highly variable prior to the V1 region, making primer selection more complex (Graspeuntner et al., 2018). These biases can be partially mitigated by recognizing the characteristics of 16S rRNA sequencing and thoughtfully selecting primers. Studies that evaluate the influence of selected 16S regions on GBS representation have not yet been published. From our findings, the 13 studies that detected GBS used 16S regions between V1–V4 or V6. However, the 32 studies that did not report GBS also used the same regions. Thus, assessment of bias by 16S region is important when designing and evaluating future vaginal microbiome studies, and further targeted studies evaluating the influence of primer region on known pregnancy pathogens should be completed.

Given limitations in exploring the true proportions within the microbial community, studies have increasingly employed whole genome sequencing (WGS), which allows for higher resolution up to the strain level. In our review, two studies employed WGS but did not report GBS (Goltsman et al., 2018; Price et al., 2019). Both papers initially selected some of the most prevalent species-level taxa, which are frequently Lactobacillus and Gardnerella species, to continue further analysis. This approach may have constrained the strength of WGS, causing researchers to miss less abundant, yet clinically important genus, species, and strains. If WGS was completed to gain a more inclusive representation of all species in the vaginal microbiome, analysis should include more taxa instead of only including the same top taxa that were included in previous 16S studies.

Limited assessment and lack of focus on low abundance organisms

The above results indicate that reporting strategies, rather than methodology, may contribute to the underreporting of less abundant pathogens in metagenomic vaginal microbiome studies during pregnancy. As the majority of studies in our review explored the compositional variation of vaginal microbiome during pregnancy, underreporting can be a serious issue because known pathogens may not be analyzed as potential contributors to pregnancy complications.

We observed differences in threshold determinants for analysis or reporting. Among the 32 studies that did not report GBS, 24 analyzed the microbiome at the species level, but only reported the most prevalent genus (i.e., Lactobacillus species). This may lead to a misclassification error, in which only the most abundant bacteria or species is associated with the outcome of interest. Additionally, 20 studies reported Streptococcus at the genus level, but did not conduct further species-level analysis. It is possible that studies that did not speciate Streptococcus had GBS present within their population, a result which could be re-evaluated if raw sequencing data is made available.

Moreover, four studies that did not report GBS in their results or microbiome data identified GBS-positive women in their study based on culture-based clinical testing (Brumbaugh et al., 2016; Jayaprakash et al., 2016; Roesch et al., 2017; Leizer et al., 2018). This suggests that there may be a benefit to using both sequence-based and culture-based methods to verify presence or absence of low abundance microbes within vaginal microbiome communities. Previous studies have shown discordance in culture-based GBS colonization results between antenatal screening and culture on admission for delivery (McCoy et al., 2020; Hussain et al., 2019). In a study by McCoy and colleagues, nearly 40% of women that were positive at GBS screening were not GBS positive at the time of admission (McCoy et al., 2020). Alternatively, Hussain et al. (2019) reported a discordance rate of 11%. While the transient nature of GBS colonization is known, factors that contribute to loss or gain of maternal GBS colonization remain uncertain (Hansen et al., 2004).

Evaluation of culture-based GBS screening previously determined that GBS transmission to neonates is more likely when maternal colonization is heavy, as determined by the number of colonies grown at the time of screening (Berardi et al., 2014; Seedat et al., 2018). However, it is unclear how well clinical culture-based microbial assessments correlate with the detection of clinically relevant pathogens in sequencing-based vaginal microbiome studies. Furthermore, much of the research investigating risk of neonatal GBS infection related to maternal bacterial load was completed in the 1980s and has not been evaluated using sequencing-based methods (Seedat et al., 2018). Future studies that concurrently evaluate the accuracy of GBS reporting within microbial communities via metagenomic sequencing approaches and culture-based methods are needed.

Strengths and limitations

Strengths of this review include the evaluation of factors that may influence the reporting of less abundance vaginal pathogens in existing microbiome literature for a known pathogen that is rarely discussed in the vaginal microbiome literature. We identified studies from multiple databases and assessed differences across studies at various decision points to analyze metagenomic data.

The heterogeneity across studies in methods and data reporting limited the ability to pool and re-analyze the data. This was further complicated by the fact that some studies reported most microbial taxa at solely the genus level, whereas other studies only included species-level data for specific genus like Lactobacillus. Furthermore, the format and availability of data associated with manuscripts also varied greatly (e.g., raw data versus including only taxonomy tables used in the analysis). In cases where the full microbial datasets were not available, it was difficult to determine whether the absence of GBS reporting was due to lack of detection, exclusion from analysis because of low abundance, or some other factor. These limitations highlight the importance of making study analysis code and raw data freely available upon publication in order to encourage reproducibility and comparison.

Conclusion

Only a small proportion of vaginal microbiome studies reported the presence of GBS during pregnancy. No systematic differences in factors previously known to introduce bias in microbiome studies were associated with the lack of GBS reporting. However, there was considerable heterogeneity in research methods employed across studies. Consideration of less abundant, but clinically meaningful microbes in vaginal microbiome studies may improve our understanding of how the vaginal microbiota influences pregnancy outcomes. GBS was reported in studies across all trimesters of pregnancy using metagenomic methods, although reporting of culture based GBS does not always correlate with metagenomic GBS reporting. Therefore, studies evaluating the presence of GBS may need to adopt both confirmatory testing for GBS via culture and parallel comparison with results obtained from metagenomic/meta-taxonomic approaches.

Supplemental Information

Supplemental Information 1 PRISMA checklist

Click here for additional data file.

Supplemental Information 2 GBS Reported Studies (n = 13)

Click here for additional data file.

Supplemental Information 3 GBS Not Reported Studies (n = 32)

Click here for additional data file.

Supplemental Information 4 SEARCH STRATEGY

Click here for additional data file.

Supplemental Information 5 Systematic Review Rationale

Click here for additional data file.

The content is solely the responsibility of the author and does not necessarily represent the official views of the NIH.

Additional Information and Declarations

Competing Interests

Author Contributions

Data Availability

The authors declare there are no competing interests.

Sungju Lim, Shilpa Rajagopal and Michelle L. Wright conceived and designed the experiments, performed the experiments, analyzed the data, prepared figures and/or tables, authored or reviewed drafts of the paper, and approved the final draft.

Ye Ryn Jeong and Dumebi Nzegwu performed the experiments, analyzed the data, prepared figures and/or tables, authored or reviewed drafts of the paper, and approved the final draft.

The following information was supplied regarding data availability:

This study is a systematic review.

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
