# Peer review of "Group B Streptococcus and the vaginal microbiome among pregnant women: a systematic review"

_PeerJ, doi:10.7717/peerj.11437_

## Round 0.1 · original submission · Minor Revisions

The manuscript will be accepted with minor revision following the suggestions by Dr. Kathryn Patras (Reviewer 2).

Reviewer 1 ·

Basic reporting

This article is well-written and clear. The background is sufficient to understand the data presented and tables and figures are clear. Supplemental tables are helpful and could even be included in the main text.

Experimental design

Experimental design is clearly laid out and appropriate. The research question is meaningful and highlights the lack of data on GBS and other potential pathogens in the vaginal microbiota as well as the problem with variability between microbiome studies. This paper brings together many relevant studies and presents a collection of data that will be useful to the GBS and vaginal microbiota scientific communities.

Validity of the findings

The rationale for why the study was done is clear and well-reasoned. This review will benefit the literature by bringing together disparate microbiota studies to examine together. Statistics and conclusions are sound and well described.

Additional comments

Overall this paper was insightful and I believe it will be a helpful addition to the scientific literature on this topic. Striving for less variability in metagenomics approaches as well as increased access to raw data will benefit the field.

A minor issue to address in editing would be to check for the capitalization and italics issues when referring to Lactobacillus as a genus.

Overall interesting and well-written article.

·

Basic reporting

The article is well-written and figures and tables are appropriately formatted.

Experimental design

Meta-analysis study design is appropriate and cohesive.

Validity of the findings

Conclusions are valid and study strengths and limitations are adequately addressed.

Additional comments

Lim et al present a new meta-analysis on the prevalence of Group B Streptococcus reporting in vaginal microbiome studies in pregnancy. GBS is an important pathogen in pregnancy and the neonatal period, however, the inconsistencies in reporting GBS status, or GBS taxonomic identification, in studies examining the role of the vaginal microbiome in pregnancy outcomes may miss the true rate of GBS-associated adverse events. The authors perform a comprehensive assessment of the current literature, using the widely accepted PRISMA guidelines. They identified 45 studies, and further stratified based on GBS reporting. They found no differences between sample sizes, methods, or data analyses. A limitation to restricting the inclusion criteria to pregnancy only is that the breadth of non-pregnant vaginal microbiome studies is missed, however, the authors provide adequate rationale for limiting this meta-analysis to pregnancy studies only. Overall, the manuscript is well-written, and the study design and rationale are excellent. My specific comments below are minor.

Minor comments:
Tables S1 and S2: How many of the studies deposited raw sequencing data? It would be useful to readers to include a column indicating studies that have publicly deposited sequencing data (and the accession numbers if applicable).
Line 76: Add a specific reference for the CDC here, or consider a recent meta-analysis of GBS colonization rates (PMID: 29117327).
Line 188: The wording of this sentence is awkward and could be interpreted that Streptococcus and Lactobacillus would be combined at the genus level. Consider re-wording/separating “as well as when studies only speciate lactobacillus” to a separate sentence.

---

## Round 0.2 · accepted · Accept

The issues are appropriately addressed.